# The Effect of *Ficus semicordata* Fig Quality on the Sex Ratio of Its Pollinating Wasp *Ceratosolen gravelyi*

Xiaoyan Yang [1,2,†], Yunfang Guan [1,2,†], Changqi Chen [1,2], Ying Zhang [1,2], Yulin Yuan [1,2], Tiantian Tang [2], Zongbo Li [2] and Yuan Zhang [1,2,*]

[1]   Yunnan Academy of Biodiversity, Southwest Forestry University, Kunming 650233, China; xiaoyan_yang-12@swfu.edu.cn (X.Y.); g_yunfang@swfu.edu.cn (Y.G.); chen_cq@swfu.edu.cn (C.C.); zhangying@swfu.edu.cn (Y.Z.); yulin_yuan@swfu.edu.cn (Y.Y.)
[2]   College of Biodiversity Conservation, Southwest Forestry University, Kunming 650233, China; tiantiantang@swfu.edu.cn (T.T.); lizb@swfu.edu.cn (Z.L.)
[*]   Correspondence: yuanzhang@swfu.edu.cn
[†]   These authors contributed equally to this work.

**Abstract:** The interaction between fig wasps and their host fig trees (*Ficus* spp.) is a striking example of an obligate pollination mutualism. Male and female fig wasps are confined within their natal patch instead of panmictic; under this circumstance, mating only occurs between individuals of the same patch. This is known as a local mate competition (LMC). It pays foundresses to invest mainly in daughters and to only produce enough sons to ensure that all female offspring can be fertilized, but in nature, pollinating fig wasps may face many problems with host quality, such as limitation of oviposition sites and the nutrition deficiency of the host fig. The sex ratio of wasps can determine the stability of fig–fig wasp mutualistic system and, thus, the stability of other species associated with it. In this study, we controlled the quality of host figs in three ways. The results showed that the host fig age can influence the sex ratio of pollinator offspring, while the foundress numbers and the presence of pollen have no significant effect on it. A compelling explanation for this result is that the sex-dependent mortality occurs. This is a novel finding of how host quality influences the interaction of fig and fig wasps, which can also help us understand the evolution and stability mechanism of this symbiotic system.

**Keywords:** symbiosis; mutualism; sex ratio; pollinator; LMC; clutch size; fig wasp





## 1. Introduction

The studies on sex ratio variation are important for understanding the causes and consequences of selective pressures that influence population structure, mating systems, and sociality [1–3]. Fig wasps (Hymenoptera, Agaonidae) have been widely utilized in studies of sex ratio evolution. Their mating is confined within their natal patch instead of panmictic; under this circumstance, mating only occurs between individuals on the same patch, sons probably have to compete with siblings for mates [4–6], so female-biased sex ratios are favored when wasps populations consist of multiple patches with limited dispersal between figs, which is known as local mate competition (LMC) [7]. When there is only one mother laying eggs, it pays the foundress to invest mainly in daughters and to only produce enough sons to ensure that all female offspring can be fertilized [8,9]. The female-biased progeny sex ratios were also explained by the theory of local resource enhancement (LRE); the beneficial effects females have on each other's reproductive success is expected to lead to female bias due to local resource enhancement [10].

Hymenopterans are ideal subjects for research on sex ratios [11], primarily due to the population structures concurring with the criteria of LMC predictions [7]. Adjustment is achieved through their haplodiploid sex determination mechanism, where unfertilized eggs produce haploid males parthenogenetically, whereas the diploid females are produced

by normally fertilized eggs. Eggs can be fertilized individually during oviposition, so each foundress can determine the sex of offspring actively [12]. The sex ratio of fig wasps can partly determine the stability of fig–fig wasp mutualism, thereby impacting the diversity and stability of other species associated with the system [6].

The predicted qualitative effects of LMC have been confirmed in many species of fig-pollinating wasps, with sex ratios becoming less female-biased as the number of ovipositing females sharing a fig increases [13,14]. However, it has been observed that the sex ratio tends to be more female-biased than predicted by extensions to LMC theory [13,15]. There are several reasons why some assumptions in basic LMC models are not met by fig–fig wasp biology [13]. For example, LMC assumes that foundresses within a patch oviposit simultaneously and have equal clutch sizes. However, oviposition by fig wasps is never truly simultaneous, but often somewhat sequential [16]. Foundresses regularly experience intraspecific competition for oviposition sites, and some foundresses lay complete egg loads while the others leave fewer eggs in multi-foundress figs [17]. Moreover, the predicted sex-ratio response for single-foundress broods is to produce a vanishingly small proportion of males, approaching zero in LMC. In fact, males are needed not only for mating but also for cutting the exit holes needed by the mated female pollinating fig wasps to leave the syconium. Therefore, at least 12% of males on average exist to ensure the reproductive success of females [13].

Host quality can be seen as a resource available to offspring, which is known to influence sex allocation in many parasitoid species where females are allocated to hosts of a higher quality [18,19]. The quality of the environment in which parents produce offspring may affect their own current and future offspring production as well as the future fitness of their offspring [20]. Host quality may influence sex ratio by the number of offspring, or by different selection when it has a differential effect upon the relative fitness of the male and female through survival and fecundity [21]. For example, a poor-quality host may result in reduced size for some parasitoid insects, and if female fitness is more adversely influenced by small size than the fitness of male, producing a larger proportion of male offspring may be favored in a poor-quality host [10]. However, for fig wasps, which are subject to intense local mate competition, the mating pattern and resource allocation are distinct from parasitoid insects and will be influenced by the brood size strategy of wasps. It is possible that the regulatory factors influencing offspring sex ratios may also be different [22].

For many parasitoid species, size of a host is an indication of quality, with larger hosts being better resourced. Similarly, for *Ficus* species, the same may be true. For instance, (1) the number of flowers correlates with fig size [23], and (2) offspring production can be significantly affected by fig size, as found in *Ficus racemosa* [24]. In fig wasps, foundresses may assess other aspects of host quality besides size. In a previous study, unpollinated figs can remain receptive to their host-specific pollinators for long periods; figs that were visited late in receptivity produced fewer, smaller wasps [25]. Hence, in addition to fig size, fig age might be another important factor to estimate host quality. According to our previous experience, in the field, heterogeneity among patches should be common because all the hosts cannot be of the same stage or quality. Therefore, it seems important to take the host quality into account, while Hamiltonian sex ratios are expected regardless of the host type in the patch. In fig–fig wasps mutualism, there are many factors that can influence the quality of a host fig, which include number of flowers in the fig, fig age [26], spatial location of the female flower [27], the season when the pollinator enters the fig [28], and fig wasp pollen-load condition [29]. These factors affect the growth and reproduction of fig wasps in two ways: one is nutrition or conditions that the fig can provide for the development of offspring, for instance, pollination may facilitate the number of wasps' offspring [30,31]; the other is the number of oviposition sites in the figs, like the later foundresses which have limited flowers for laying eggs [27].

The quality of the host is important, not only because it affects suitability for oviposition, but also because it impacts the size and quality of the clutch that it can support [32]. Furthermore, host quality may influence the mutualistic relationship through the variance

of wasps sex ratio. If fig quality can really influence wasp sex ratio, the factor should be considered when we are testing how some other factors impact the sex ratio, and experiments should be designed to rule out the effects of fig quality. In this study, we conducted controlled experiments from 3 aspects which can possibly influence the quality of a host fig in *F. semicordata*. The following questions will be answered: (i) How did host age affect the wasps' sex ratio? (ii) How did foundresses entering sequences influence the wasps' sex ratio when the effect of fig age was controlled for? (iii) Is there any difference in wasps' sex ratio between foundresses that were pollen-loaded (P+) and those that were pollen-free (P−)?

## 2. Materials and Methods

### 2.1. Study Area

The experiment was conducted in the Xishuangbanna Tropical Botanical Garden, Chinese Academy of Sciences, Yunnan, China, Southeast Asia, located at 21°41_N, 101°25_E, at an altitude of approximately 600 m. The climate of Xishuangbanna is dominated by southwest monsoon with intense rainfall occurring from May to October. Therefore, the climate is characterized by dry, rainy, and foggy seasons, which last from March to May, June to October, and November to February, respectively. The average relative humidity of Xishuangbanna is 86%, and the average temperature is 21.4–22.6 °C. For our experiments, two trees were selected to conduct these controlled experiments. One tree was used for foundress collection, another tree for the experiments.

### 2.2. Pollinating Fig Wasp Reproductive Biology

The interaction between fig wasps and their host fig trees (*Ficus* spp.) is a striking example of an obligate pollination mutualism [5]. The fig is urn-shaped and lined with numerous tiny uniovulate female flowers. Species-specific fig wasp females (Agaonidae: Hymenoptera) carry pollen from natal figs into a receptive fig, entering through a narrow bract-lined ostiole. Once inside, the foundresses actively or passively pollinate the flowers, some of which can also be galled and have eggs laid in them [5,33]. After entry by foundresses, the figs and the wasp offspring reach maturity after several weeks, which is the male phase. All wasp reproductive activities occur within the confines of the fig. Foundresses determine the sex of offspring because unfertilized eggs produce haploid males, and the diploid females originate from normally fertilized eggs. Eggs can be fertilized individually during oviposition. When the wasps reach maturity, the male wasps emerge from their galls first and seek galls containing female wasps, then males chew holes in the gall of females and mate with females though the hole. After mating, females exit the gall and actively or passively collect the pollen inside the natal fig, then females leave figs through the tunnel created by the males and fly in search of receptive figs, which are usually on another tree [5,34]. Thus, the fig wasp system gives added selective pressure to pollinating fig wasp foundresses to produce "insurance males" [35]. Whether the wasps oviposit depends on the breeding system of the host *Ficus*. If it is a monoecious species, then each syconium produces seeds and pollinator offspring, but in dioecious fig trees, there is reproductive specialization, with male trees producing only pollinator offspring and no seeds, and female trees producing only seeds [33] (Figure 1).

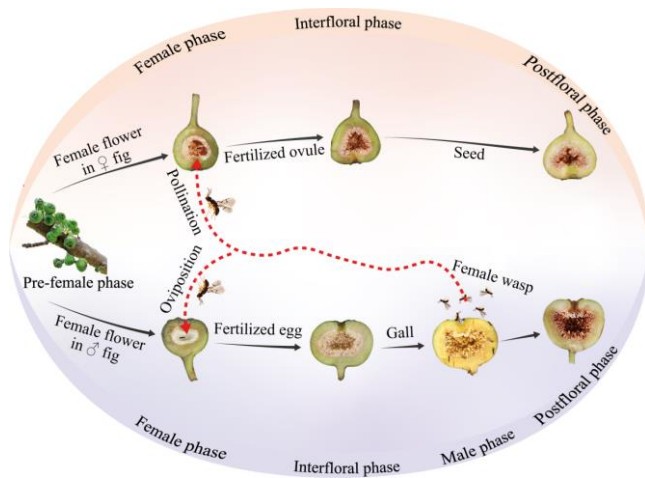

**Figure 1.** Life history for dioecious figs.

*2.3. Study Species*

We conducted experiments on *F. semicordata*, a dioecious fig species with male figs producing wasps and female figs producing seeds only. The tree grows to a height of approximately 3–10 m and produces its figs on long, slender branches that hang down near the ground. Figs of *F. semicordata* are produced in two or three synchronous crops annually, with different trees fruiting at different times. The diameter of figs on the first day of receptivity varies from 12 to 16 mm. One female fig contains 1573 ± 14 female flowers (Mean ± SE, n = 133). Under natural conditions, the average foundress numbers are 1.67 ± 0.08 (Mean ± SE, n = 182) for each male fig. The main reason we chose this mutualistic partner is that they are easily available, and the location of figs makes experimental manipulation easy, noticeably for bagging experiments (Figure 2A). *Ceratosolen gravelyi* Grandi (Agaonidae, Hymenoptera) is the exclusive pollinator for *F. semicordata* (Figure 2B). *F. semicordata* have two types of figs at XTBG; the figs used for our experiment are the form with big figs (Figure 2C).

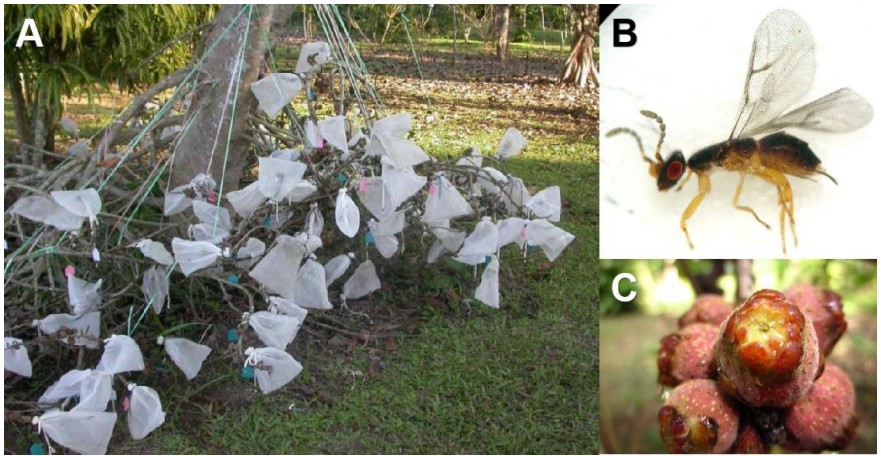

**Figure 2.** (**A**) The field experimental tree; (**B**) the pollinator *Ceratosolen gravelyi*; (**C**) the experimental figs of *Ficus semicordata*.

*2.4. The Effect of Host Fig Age on the Sex Ratio of Fig Wasp Progeny*

For testing of how fig age affects the sex ratio of wasp progeny, we enclosed pre-receptive figs in netting bags to prevent foundress entry. The onset of receptivity in *F. semicordata* was determined by the pollinator behavior: if freshly emerged pollinating fig wasps placed on the fig succeeded in partially gaining entry into the fig within five minutes (at which point they were removed), the fig was deemed receptive. Pre-receptive figs were

tested on successive days. Once the first day of receptivity was determined, the fig was labelled and re-enclosed with a netting bag. A single pollinator was then allowed to enter individual figs of known advancement in receptivity. Pollinators were introduced into figs on the 1st, 2nd, 3rd, 4th, 5th, 6th, 8th, and 12th days after the figs became receptive. After entry by pollinators, the figs were bagged again to prevent further pollinator entries or parasitism by non-pollinating fig wasps and left to develop. The figs that reached maturity were removed from the branches just before wasp emergence and placed into netting bags. The pollinator progeny was allowed to emerge naturally, and then the female and male progeny numbers for each fig were counted separately. Wasps that failed to emerge were also counted by dissecting the fig. The sex ratio of fig wasp progeny was calculated as the ratio of the number of males divided by the number of total offspring, the same as below.

### 2.5. The Effect of Foundresses Entering Intervals on the Sex Ratio of Fig Wasp Progeny

To examine how the time interval separating foundress entries into figs affects the sex ratio of wasp progeny, we introduced pollinators into figs at different time intervals. First, we established the first day of figs receptivity as above. Then, a pair of foundresses were introduced in figs in two different ways: (1) one foundress was deposited on the fig on the first day of receptivity, and after it had entered the fig, a second foundress was immediately placed on the fig, the fig was enclosed again in a netting bag after successful entry into the fig of the second foundress; (2) alternatively, one foundress was introduced into a fig on their first day of receptivity, and the bag was replaced as before, the second foundress was allowed to enter the fig the following day, and then the bags were replaced.

After several weeks, nearly ripe figs were gathered, and the numbers of female and male progeny for each fig were counted separately, as in the previous experiments. We carried out this experiment with 2 treatments, that of 2 (24 h interval for the second foundress) and 3 foundresses (24 h for the second foundress and 48 h for the third foundress). In our 24 h interval treatment, figs losing receptivity in the second or the third day of receptivity were discarded.

### 2.6. The Effect of Pollen on the Sex Ratio of Fig Wasp Progeny

We also tested the wasp progeny sex ratio changing with host quality from another perspective, that is, foundresses carrying pollen or not. First of all, the onset of receptivity was detected as above, then we experimentally produced pollen-carrying and artificially pollen-free wasps. To obtain artificially pollen-free wasps of the pollinator species, we gathered nearly ripe figs and opened the figs when male wasps were mating with the females, but when females were still within their galls, all male flowers were removed with forceps lightly to prevent female wasps from accessing pollen when they emerged. Wasps carrying pollen were obtained by letting them emerge naturally from mature figs into netting bags. When figs on the experimental tree were at the first receptive day, a single pollen-carrying or pollen-free female wasp was introduced into each fig, then the bags were replaced, and the numbers of male and female wasps' offspring were collected and counted separately for each ripe fig as described above.

### 2.7. Statistical Analysis

The relationship between fig age and offspring numbers/sex ratio was analyzed using generalized linear model (GLM). Offspring numbers and sex ratios were the response variable, whereas fig age was the explanatory variable. We assumed Poisson error variance for offspring number and binomial error variance for sex ratio. To test for the effect of the foundresses entering interval as well as the presence of pollen on offspring numbers and sex ratio, one-way ANOVA was used. To homogenize variances, data were log-transformed when needed. The analyses were carried out using R 4.2.2.

## 3. Results

### 3.1. The Effect of Host Fig Age on Sex Ratio of Wasp Progeny

When no fig wasps were allowed to enter the figs, they remained receptive for almost 2 weeks. Our results showed that when the length of figs waiting for pollinating fig wasps increased, the offspring number decreased significantly (GLM: Slope = $\beta \pm SE = -0.04 \pm 0.002$; $p < 0.001$); when the response variables are considered as female and male progeny separately, the results indicated that the age showed significant influence on both male and female offsprings, while a larger coefficient was found in males (GLM for females: Slope = $\beta \pm SE = -0.04 \pm 0.002$, $p < 0.001$; GLM for males: Slope = $\beta \pm SE = -0.07 \pm 0.005$, $p < 0.001$). Similarly, fig age also presented a significant inverse association with sex ratio (GLM: Slope = $\beta \pm SE = -0.027 \pm 0.005$; $p < 0.001$) (Figure 3).

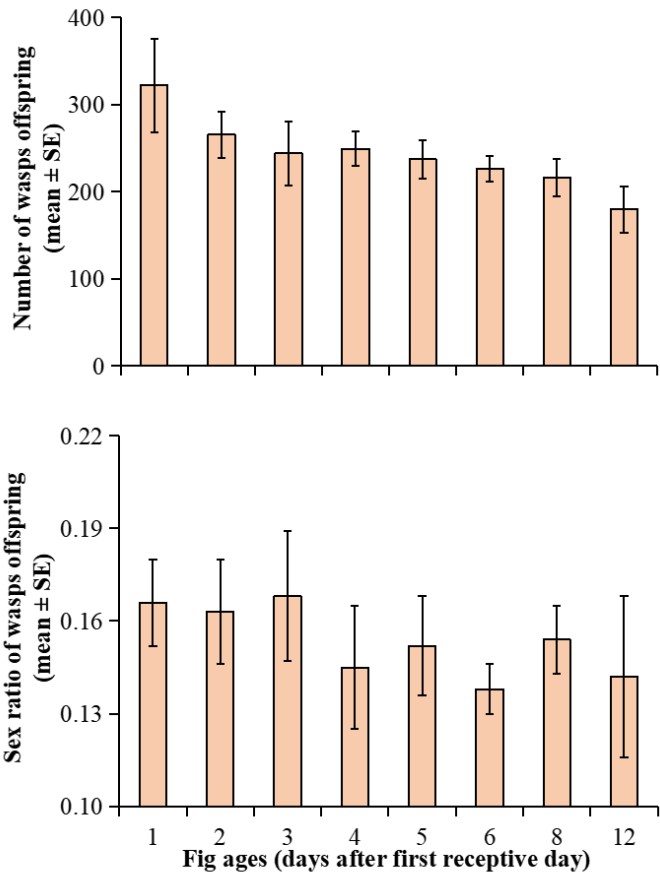

**Figure 3.** Offspring numbers and sex ratio of pollinator progeny for different fig ages (since start of first receptive day). Sample size (figs numbers) = 18, 18, 15, 18, 20, 16, 17, 20 for experiments of fig ages = 1, 2, 3, 4, 5, 6, 8, 13, respectively.

### 3.2. The Effect of Foundresses Entering Intervals on Sex Ratio of Wasp Progeny

For two foundresses treatment, the average offspring numbers of foundresses entering figs successively tended to be higher (F = 2.11; df = 1, 38; $p = 0.152$), but the progeny sex ratio tended to be lower (F = 0.023; df = 1, 38; $p = 0.880$). Somehow, the differences were not significant. For the 3 foundresses treatment, the results were similar. In the 24 h and 48 h intervals between entry experiments, the average offspring numbers tended to be lower (F = 0.09; df = 1, 36; $p = 0.775$), while the progeny tended to be less female-biased, but the difference was not significant either (F = 0.396; df = 1, 36; $p = 0.396$). When only one foundress had been introduced, the mean receptive periods were $2.17 \pm 0.15$ days (Mean $\pm$ SE), with 41% and 80% figs losing receptivity for the second and the third day, respectively (Figure 4).

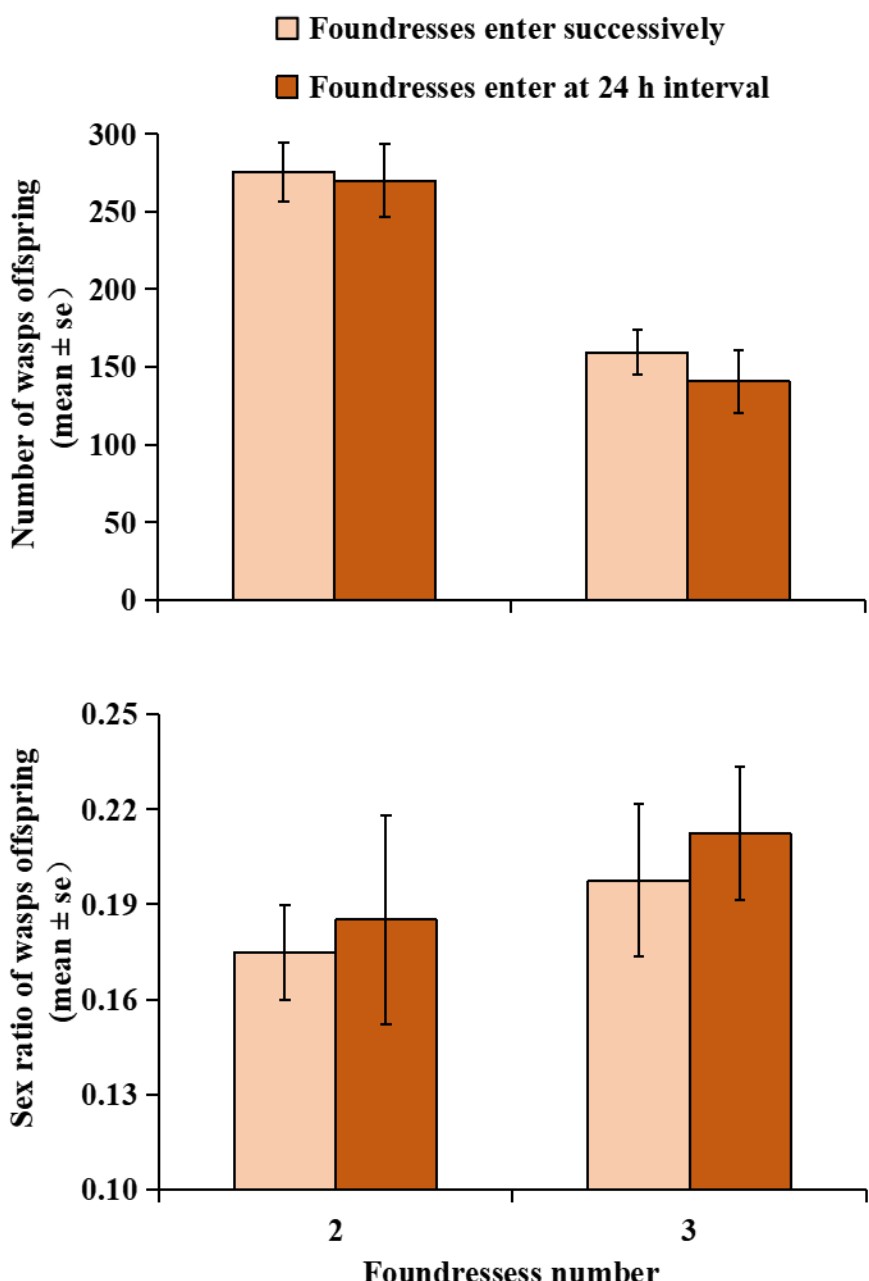

**Figure 4.** Offspring numbers and sex ratio of pollinator progeny for different ways that pollinators enter into figs. Sample size = 21, 19 for experiments of 2 foundresses, 20, 18 for experiments of 3 foundresses.

### 3.3. The Effect of Pollen on Sex Ratio of Wasp Progeny

For pollen exclusion experiment, the results showed that the presence of pollen resulted in a significantly larger offspring number (F = 12.41; df =1, 43; $p < 0.01$), and progeny sex ratio tended to increase, but the results were not significantly different (F = 2.61; df =1, 43; $p = 0.113$) (Figure 5).

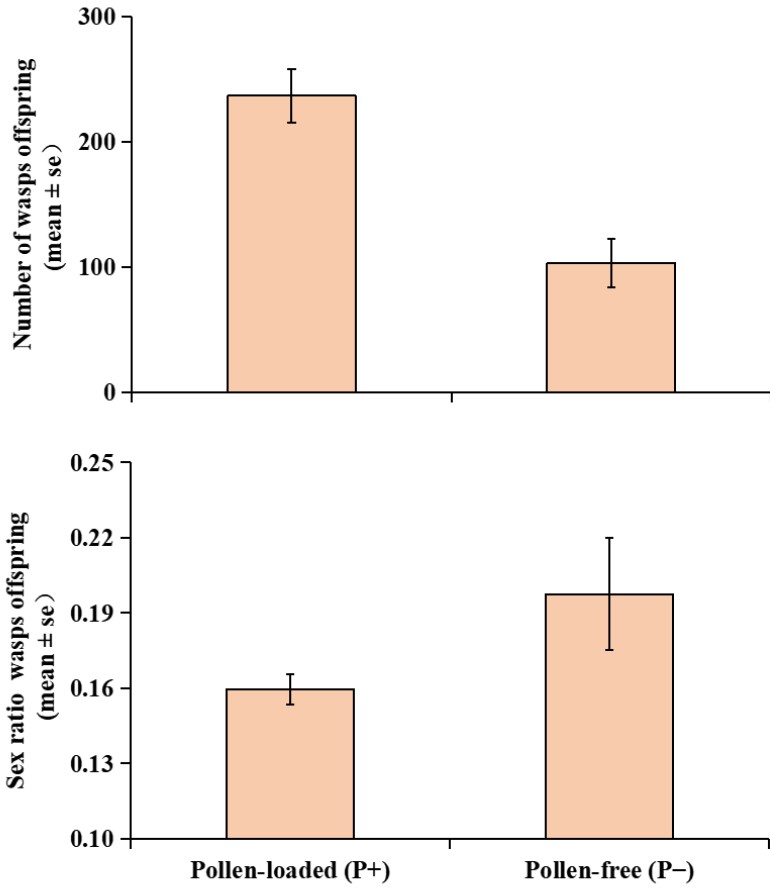

**Figure 5.** Offspring numbers and sex ratio of pollinator progeny for foundresses with and without pollen. Sample size = 33, 12 for experiment of P+ and P−, respectively.

## 4. Discussion

*Ceratosolen gravelyi*, like most of Hymenoptera, is haplodiploid, with unfertilized eggs developing into males and fertilized eggs into females. This probably means that the sex ratio of offspring could be partly regulated by the foundress laying different types of eggs [9]. That is the reason sex ratio has been especially well studied in Hymenoptera. It has been believed that foundresses can adjust the sex ratio of their offspring to maximize their biological fitness since the work by Hamilton in 1967 [7]. However, the meta-analysis by West and Sheldon indicated that the evidence of such environment-oriented sex adjustment is not always convincing [11,13,36]. In our experiment, three different factors which may affect the quality of host figs, and thus the sex ratio of pollinating fig wasps, have been researched. First of all, regarding the effect of fig age on wasps, the offspring sex ratio was examined, and the results showed that the sex ratio of wasp progeny in older host figs tended to be more female-biased. At the same time, clutch size of wasp offspring developing in aged figs was significantly decreased, so sex ratio was positively associated with clutch size in this experiment. This result was not consistent with the prediction of LMC, which predicts that in single-foundress figs, a higher proportion of males should be laid when clutch size is small [15]. This inconsistency is quite possibly due to gender-biased mortality, and our results showed that male offspring have higher mortality. This is the first experiment on fig wasps to examine how host fig age influences the sex ratio of wasps' offspring. Most of the studies on parasitoids differ from the system of figs and fig wasps, in that the relationship of those two species is in a mutualism rather than a parasitism. Furthermore, we have found that wasps developing in older figs have higher mortality (estimated by aborted gall) [25], which is probably because of wasps developing in aged figs experiencing stronger resource or nourishment competition, or partly because wasp

larvae had failed to complete their development. In the fig wasp system, male flowers develop at a fixed time, so developmental time has to be shortened for wasps' entry into aged figs to synchronize the mature time for both wasps' offspring and fig anthers. We hypothesize that haploid male wasps might be more sensitive to poor environments, or that they have a weaker ability for resource competition under poor resource conditions. For now, we are unable to identify the developmental mortality for both sexes since the sex of larvae within an aborted gall cannot be determined, especially for galls aborted in the early developmental stages. Differential developmental mortality between parasitoid sexes has been explored in very few cases, and no significant difference has been shown [37]. Even so, our study corroborated the findings on another *Ceratosolen* species and another non-fig wasp that males die more easily [37,38]. As we know, fig and fig wasps depend completely on each other for reproduction, and any character benefit for both species should be favored. It will be to the benefit of both fig and pollinators if a higher proportion of females can be produced, especially under the circumstance of clutch reduction. The sex ratio decreased with fig age, which means that female offspring do not decline as significantly as male offspring. It might be a compensation effect for both aged fig and fig wasps entering them.

In this study, we also demonstrated the effect of host quality on offspring sex ratio from another perspective by introducing the foundresses that were pollen-loaded or pollen-free. Previous studies have suggested that active pollinators efficiently fertilize flowers in which they oviposit, and wasps' larvae feed on the nucellar tissue of normally double-pollinated flowers; thus, a lack of pollination increases larval mortality [6,39], and active pollination has probably evolved as a way to improve progeny nourishment [29,40,41]. This idea is supported by larger broods in pollinated *F. hispida* [31]. According to LMC theory, the decrease in the number of offspring leads to an increase in the sex ratio, while in our experiments, we found an increase but non-significant difference between the offspring sex ratio of pollen-loaded foundresses and that of pollen-free foundresses, so we infer that when nourishment competition is more intense, the haploid male offspring may have a higher mortality rate, the similar theoretical assumption with fig-age-related experiment.

In addition to the above two factors, fig wasps may often face another problem of host quality in nature, which is the limitation of oviposition sites. According to our previous research, a single pollinator can use about a third of the flowers [28], which means that even the number of foundresses reaches more than two; the offspring number does not increase significantly even as the number of pollinators increases, so we designed the experiment to make the second entry wasps have less flowers to oviposit, as less space for progeny to develop. Our result showed that the sex ratio was higher although a non-significant difference was shown when foundresses enter with 24 h interval, and the results were similar for both two and three wasps experiment. Our results show similarities with some previous studies. This implies that foundresses may not adjust the sex of their offspring when the host quality declines [36]. That may be because under the circumstance of a longer interval, the second one will not be able to lay her full egg load due to oviposition site limitation or the loss of receptivity. Furthermore, fig wasps first lay a set number of male eggs and then female eggs, or at least mainly male eggs initially [14,39]. We infer that oviposition site limitation would be the reason for sex ratio increasing rather than that the second foundress can perceive the existence of the first foundress according to previous studies [42], so the later foundress probably cannot access the full capacity of oviposition due to the age of female flower changing. The offspring of a foundress entering at different times may have different mortality; however, we cannot separately count the progeny sex ratio for every single foundress when more than one foundress enters the fig. It is necessary to conduct more experiments to figure out the mechanism.

In this study, we carried out our experiments uniformly on the first day of fig receptivity to rule out the possible effect of receptivity. This design is an important supplement to previous research, and makes the experiment more difficult, but it is scientifically necessary, especially under the condition that fig age has an effect on sex ratio of wasps' offspring. Our study revealed how sex ratio and clutch size respond to host quality from three differ-

ent perspectives. Our findings also provide scientific evidence for subsequent controlled experiments in fig–fig wasps. Our results indicated that fig–fig wasps may exhibit characteristics distinct from many other insect species. In certain instances, the sex ratio of fig wasps cannot be fully predicted by the LMC model. This suggests the existence of many previously unidentified mechanisms influencing the sex ratio of fig wasps, which in turn affects the stability of this mutualistic system. An important question which arises from our results is what mechanism is affecting the sex ratio of wasp offspring developing in different host quality. Determining the mechanism might be the next step to understand sex ratio allocation strategies for different quality hosts [16]. To decide whether this pattern is typical across the fig species, more fig species need to be researched.

**Author Contributions:** Conceptualization, Y.Z. (Yuan Zhang) and X.Y.; methodology, Y.Z. (Yuan Zhang); software, Z.L.; validation, Y.G., C.C. and Y.Z. (Ying Zhang); formal analysis, T.T.; investigation, X.Y.; resources, C.C.; data curation, Y.Y. and Z.L.; writing—original draft preparation, X.Y.; writing—review and editing, Y.Z. (Yuan Zhang); visualization, X.Y. All authors have read and agreed to the published version of the manuscript.

**Funding:** This work was funded by grants from the National Natural Science Foundation of China (Nos. 32160296, 32260719, 31560116) and the Fundamental Research Program of Yunnan Province, China (202401AT070265). We are also grateful for the support of Joint Agricultural Project of Yunnan Province, China (202401BD070001-111) and The Young Top-Notch Talent of Yunnan Outstanding Talent Program (XDYC-QNRC-2022-0207).

**Data Availability Statement:** The original contributions presented in the study are included in the article, further inquiries can be directed to the corresponding author.

**Conflicts of Interest:** The authors declare no conflicts of interest.

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
