# Peer review of "The Effect of Ficus semicordata Fig Quality on the Sex Ratio of Its Pollinating Wasp Ceratosolen gravelyi"

_diversity, doi:10.3390/d16050298_

Round 1

Reviewer 1 Report

Comments and Suggestions for Authors

This is a very thorough study that left no stone unturned in the execution of the wet experiment. However, the analyses can be improved a lot to increase the amount that can be learnt from these data. The writing can also be improved by a better appreciation of the current literature. The language in the manuscript can also be improved significantly. I have annotated the PDF to help with that and urge the authors not to write such long sentences.

In terms of literature that is relevant I have made suggestions in the PDF comments.

With regards analyses, I suggest that the authors analyse the number of male and female offspring separately with GLM's using Poisson errors. More details are given in PDF comments. This will give far more power for analyses and more of your results may be significant. By analysing the number of male and females offspring separately, the authors will be able to address their idea that males are more likely to die directly, rather than waving their hands in the discussion.

I copy a few comments from the discussion here as I think they need special attention.

L263 It should also be discussed that this observation is at odds with the observation of Kjellberg et al. (2005) Clutch size: a major sex ratio determinant in fig pollinating wasps? Comptes Rendus Biologies 328, 471-476. where the opposite was documented in many wasps. By referring to this and analysing the number of males and females separately the authors can make a much stronger claim that this observation is the result of males dying more readily.

L265-267 This passage is very poorly constructed and what I can make out does not make sense. You cannot conclude that this is the result of mothers laying a different sex ratio. Is it not the result of males dying more readily? Such a claim would be consistent with your claims elsewhere. Separate analysis of male and female numbers are required to make this clear.

L273 supports the interpretation that it is due to higher male mortality but no data is given?

GLM's are much more powerful to detect differences. I suggest strongly that it must also be used for the latter two experiments. Only if these are also not significant should the authors accept the current non-significant conclusions.

Comments on the Quality of English Language

I made many small suggestions. On the whole the extremely long sentences must be broken up. 

Reviewer 2 Report

Comments and Suggestions for Authors

Dear Dr. Zhang,

I have carefully read your manuscript entitled "The effect of Ficus semicordata fig quality on the sex ratio of its pollinating wasp Ceratosolen gravelyi". I believe the paper contains new information on biology of C. gravelyi, and therefore it could be published in Diversity. However, I suggest certain corrections to the text (please see the attached file).

Yours sincerely,

Comments on the Quality of English Language

Dear Colleagues,

I think the quality of English is generally acceptable. However, I suggest certain corrections to the text (please see the attached file).

Round 2

Reviewer 1 Report

Comments and Suggestions for Authors

see attached file

Comments on the Quality of English Language

see attached file

Author Response

The manuscript was much improved by the corrections and I have only four questions and many more small grammatical suggestions.

Response: Thank you for your positive evaluation on our manuscript. We have carefully addressed all the comments you raised in the attached file. You may find corresponding revisions in track changes in our submitted revision. We believe our revised manuscript has been significantly improved under your help.

L212 Surely the important number of flowers is in male figs (where wasp can lay eggs as was studied here) and not in female figs. Is this just a typo? Otherwise it should be replaced.

Response: Thank you for your important comment. Exactly as you pointed out, the important number of flowers is only in male figs. Here in the sentence “the numbers of male and female progeny were collected”, we were referring to male and female wasps offspring, we calculated sex ratio of wasps offspring by counting the number of male and female progeny separately. To avoid misunderstanding, we have changed “male and female progeny” to “male and female wasps offspring”.

Figures 4 and 5 suggest that males survive better than females? This contradicts the claim that males survive less well? Although, the results are not significant. If I understand this correctly, please comment on this.

Response: Thank you for your professional comment. Your understanding is correct. Considering this result might be a little confusing to understand, here we further explain it: according to the theory of LMC, when the number of offspring decreases, the sex ratio of offspring should increase, but our results implies that host quality can bring about a significant decrease in the wasps’ offspring, while the sex ratio did not show significant decrease, so we analyze the reason for the discrepancy between our results and LMC prediction.

L378 [37] definitely found that males died more readily. So should the sentences not rather read: "Even so, our study corroborated the findings on another Ceratosolen species [37] and another nonfig wasp [38] that males die more easily."

Response: Thank you for this comment. We have changed this sentence based on your suggestion.

The data should really be made available on publication of the paper?

Response: Thank you for this important comment. Yes, for our readers to better corroborate all the results, we intend to make our data available online on the publication of our paper.

change to is indicated by an → and look carefully because a number of instances have more than one (up to 3) corrections.

L79 → simultaneously and have

L86 12%, not 5%

L97 → may be favored in a

L113 the quality of a

L114→the season when the pollinator enter the fig

L146 →quality of host figs

L159 →Therefore,

L143 → wasps' sex ratio

L162 →foundress collection

L179 leave figs through the tunnel

L181 →oviposit, depends on the

L213 → average foundress numbers

L217 →is the exclusive pollinator of F. semicordata (Fig. 2B). Ficus semicordata

L240 become→became

L279→offspring numbers and sex ratios were the response variables.

L282→as well as the presence

L283 → numbers and sex ratio

L307 → 6, 8 and 12 respectively.

L346 →wasps'

L371 →In the fig-wasp system

L373 → both wasps' offspring and fig anthers

L374 → or that the have

L391 →, thus, a lack

L392 → This idea is supported by larger broods in pollinated F. hispida [31]

L397 →pollen-laiden

L398 → may have a higher

L402 potentially insert the reference to your previous research.

L405 →wasp have less flowers to oviposit into

L408→experiments L408-410 [16] does not report any sex ratio. In your original submission you referred to a different paper.

L410 →That may be because

L410 → of a longer interval

L411 → to lay her full egg load

L414 → rather than that the second

L416 → cannot access the full capacity of oviposition

L490 → enter the fig. It it

L495 →fig age has an effect

L496→size respond to

L502 →consider citing someone for the new text?

Response: Thank you so much for this detailed comment. We have fully addressed all the errors you mentioned. Please see in track changes in the submitted revised manuscript.
